# The Chemical Composition and Nutritional Value of Chia Seeds—Current State of Knowledge

**DOI:** 10.3390/nu11061242

**Published:** 2019-05-31

**Authors:** Bartosz Kulczyński, Joanna Kobus-Cisowska, Maciej Taczanowski, Dominik Kmiecik, Anna Gramza-Michałowska

**Affiliations:** 1Department of Gastronomy Sciences and Functional Foods, Faculty of Food Science and Nutrition, Poznań University of Life Sciences, Wojska Polskiego 31, 60–624 Poznań, Poland; bartosz.kulczynski@up.poznan.pl (B.K.); joanna.kobus-cisowska@up.poznan.pl (J.K.-C.); dominik.kmiecik@up.poznan.pl (D.K.); 2Department of Food Quality and Management, Faculty of Food Science and Nutrition, Poznań University of Life Sciences, Wojska Polskiego 31, 60–624 Poznań, Poland; maciej.taczanowski@up.poznan.pl

**Keywords:** *Salvia hispanica*, chia seeds, fatty acids, omega-3, antioxidant activity, health-promoting properties

## Abstract

Chia (*Salvia hispanica*) is an annual herbaceous plant, the seeds of which were consumed already thousands of years ago. Current research results indicate a high nutritive value for chia seeds and confirm their extensive health-promoting properties. Research indicates that components of chia seeds are ascribed a beneficial effect on the improvement of the blood lipid profile, through their hypotensive, hypoglycaemic, antimicrobial and immunostimulatory effects. This article provides a review of the most important information concerning the potential application of chia seeds in food production. The chemical composition of chia seeds is presented and the effect of their consumption on human health is discussed. Technological properties of chia seeds are shown and current legal regulations concerning their potential use in the food industry are presented.

## 1. Introduction

Adequate nutrition is an important element in the prevention of many civilisation-related diseases such as diabetes, cardiovascular disease and obesity. Both state institutions and non-governmental organisations issue nutritional recommendations to protect human health, inhibit the development of selected diseases and alleviate their symptoms [1]. An increasingly important health-promoting role is ascribed to bioactive food components. They were defined by Biesalski and co-workers as nutritional components or non-nutritional compounds naturally found in the raw material or formed in the product in the course of technological processes, which may enhance, inhibit or modify physiological and metabolic functions of the organism [2]. The American Dietetic Association supplements this definition by additionally stressing the importance of health safety of bioactive food [3]. Bioactive compounds include, e.g., polyphenols, carotenoids, phytoestrogens, sterols, stanols, vitamins, dietary fibre, fatty acids, probiotics, prebiotics, and bioactive peptides [4,5,6]. In view of the health-promoting properties of food in recent years we have been observing a considerable interest in products of plant origin, which have been investigated in many studies [7,8,9,10]. An example of a raw material with properties considered very interesting by dietitians and food technologists is *Salvia hispanica*, commonly called chia. The word “chia” is a Spanish adaptation of “chian” or “chien”, originating from Nahuati and meaning “oily”. Chia is an herbaceous plant that has also been used for medicinal purposes for thousands of years [11,12,13,14]. Currently, chia seeds are consumed as ingredients or additions to many foodstuffs: baked products, muesli, dairy drinks, fruit smoothies or salads [15,16,17,18]. They are also used as thickeners in soups and sauces. The aim of this article is to present current information concerning potential use of chia seeds in the food industry, focusing on their chemical composition and health-promoting properties as well as legal acts regulating their use in food production. 

## 2. Botanical Characteristics

*Salvia hispanica*, also called chia, is an annual herbaceous plant belonging to the family *Lamiaceae*. This plant may reach 1 m in height. Its serrated leaves, arranged opposite, are 4–8 cm in length and 3–5 cm in width [14]. Its white or blue flowers are bisexual, of 3–4 mm in size, growing in whorls at shoot tips. After overblowing chia forms round fruits, containing many tiny, oval seeds of 2 mm in length and 1 mm in width. Seed surface is smooth, shiny, ranging in colour from white through grey to brown, with irregularly arranged black spots [14,19]. Initially, chia was grown in tropical and subtropical climates. At present, it is grown worldwide, particularly in Argentina, Peru, Paraguay, Ecuador, Mexico, Nicaragua, Bolivia, Guatemala and Australia. In Europe, it is grown in greenhouses [12,14,20]. Chia is not frost-resistant. In nature, it grows mainly in mountainous regions [12,21]. *Salvia hispanica* develops itself properly in sandy loam and clay loam soils with good drainage conditions [22]. The reported seed yield from selected commercial fields located in Argentina and Colombia ranges from 450 to 1250 kg/ha; however, under advantageous experimental conditions, the yield may rich well above 2000 kg/ha [23].

## 3. Chemical Composition

The chemical composition of chia seeds has been analysed in many studies. Detailed data on basic chia seeds composition is presented in Figure 1. Chia seeds are ascribed high nutritive value particularly thanks to their high contents of dietary fibre and fat (Table 1). Chia seeds contain approximately 30–34 g dietary fibre, of which the insoluble fraction (IDF) accounts for approximately 85–93%, while soluble dietary fibre (SDF) is approximately 7–15% [24,25]. In terms of dietary fibre content, chia seeds exceed dried fruits, cereals or nuts (Figure 2). The fatty acid profile is of particular interest. It is characterised by high contents of polyunsaturated fatty acids, mainly α-linolenic acid (ALA), which accounts for approximately 60% all fatty acids. Linoleic, oleic and palmitic acids are found in lower amounts (Table 2). Chia seeds have greater contents of omega-3 acids than flaxseed. We also need to stress the advantageous ratio of omega-6 to omega-3 acids, which is approximately 0.3:0.35 [26,27,28,29,30,31]. Chia seeds are also a good source of plant protein, which accounts for approximately 18–24% their mass [32]. Analyses of the amino acid composition (Table 3) confirmed the presence of 10 exogenous amino acids, among which the greatest contents were for arginine, leucine, phenylalanine, valine and lysine. Proteins in chia seeds are also rich in endogenous amino acids, mainly glutamic and aspartic acids, alanine, serine and glycine [27,33,34]. It needs to be stressed that chia seeds are gluten-free and as such may be consumed by celiac patients [14]. Moreover, chia seeds supply many minerals, with phosphorus (860–919 mg/100 g), calcium (456–631 mg/100 g), potassium (407–726 mg/100g) and magnesium (335–449 mg/100 g) found in greatest amounts [33,35]. Studies also confirmed the presence of some vitamins, mainly vitamin B1 (0.6 mg/100 g), vitamin B2 (0.2 mg/100 g) and niacin (8.8 mg/100 g) [33,35]. Chia seeds are also a rich source of particularly interesting groups of phytocompounds characterised by high biological activity [36,37]. These are particularly polyphenols: gallic, caffeic, chlorogenic, cinnamic and ferulic acids, quercetin, kaempferol, epicatechin, rutin, apigenin and p-coumaric acid. Isoflavones, such as daidzein, glycitein, genistein and genistin, are found in small amounts (Table 4). Ciftci et al. showed the presence of campesterol (472 mg/kg of lipids), stigmasterol (1248 mg/kg of lipids), β-sitosterol (2057 mg/kg of lipids) and Δ5-avenasterol [26]. Moreover, it was found that chia seeds also contain tocopherols: α-tocopherol (8 mg/kg of lipids), γ-tocopherol (422 mg/kg of lipids) and δ-tocopherol (15 mg/kg of lipids).

## 4. Health-Promoting Properties

Even though literature sources worldwide have presented numerous studies showing high biological activity and many health-promoting properties of chia seeds for a quite a long time, a sharp rise of interest in commercial applications of the seeds has only been observed recently [40,41,42,43]. 

Marineli et al. [44] aimed to assess the effect of consumed chia seeds on selected carbohydrate metabolism indexes. They showed that rats consuming high-fat and high-fructose diet, in which soybean oil was replaced with a 13.3% addition of chia seeds (w/w), were characterised by greater tolerance of both glucose and insulin in comparison to the control. This effect was observed during both short-term (six-week) and long-term (12-week) dietary interventions. In that study, a reduced blood concentration of non-esterified fatty acids (NEFA) is reported for the group of animals consuming chia seeds. Moreover, they recorded a decrease in the levels of hepatocellular damage markers, i.e., alanine transaminase (ALT) and aspartate transaminase (AST), the high concentrations of which were caused by the high-fat and high-fructose diet [44]. The effect of chia seeds consumption on carbohydrate and lipid metabolism was also investigated by Silva et al. [45]. In their experiment Wistar rats were divided into six groups: (1) fed casein as a source of protein; (2) fed a protein-free diet; and (3–6) receiving chia seeds or chia seed flour, with or without thermal processing. They found that the groups of animals receiving chia seeds and flour had lower blood concentrations of triglycerides (TG), total cholesterol (TC), low density lipoproteins (LDL) and very low density lipoproteins (VLDL) in comparison to the control, consuming casein. Furthermore, an increased concentration of high density lipoproteins (HDL) was recorded. They also confirmed the hypoglycaemic effect of chia. It was found that administration of chia seeds and flour reduced blood glucose level in comparison to the control [45]. Ho et al. [46] observed that individuals consuming bread fortified with chia seeds had lower postprandial glycemia in comparison to individuals, who consumed bread free from that additive. This effect was dose-dependent. The lowest level of glycemia was recorded at the addition of 24 g chia seeds, while it was highest at the addition of 7 g. Those authors suggested that the hypoglycaemic effect of chia seeds results from their high content of dietary fibre [46]. Studies conducted with type 2 diabetes patients showed that the daily supply of 15 g/1000 kcal chia seeds for 12 weeks caused a statistically significant reduction in the concentration of high-sensitivity C-reactive protein (hs-CRP) (by 40%) and von Willebrand factor (by 21%). Systolic blood pressure (SBP) was reduced by 6.3 mm Hg. No statistically significant differences were observed in blood glucose levels or parameters of the blood lipid profile (TC, LDL, HDL, and TG) [47]. 

Other researchers investigated the influence of chia oil supplementation on body composition and insulin signalling in skeletal muscles of obese mice [48]. The results indicate that mice treated with chia oil show reduced fat mass accumulation and increased lean mass, improved glucose levels and insulin tolerance, and increased blood levels of high density lipoprotein cholesterol. Creus et al. [49] reported that dietary chia seeds improve the altered metabolic fate of glucose and reduce increased collagen deposition in the heart of dyslipidaemic insulin-resistant rats fed a sucrose-rich diet.

The advantageous effect of chia seeds on the blood lipid profile was shown in an experiment conducted by Chicco et al. [50]. They observed that rats fed a diet with high sucrose contents and containing 2.6% chia seeds had lower blood concentrations of triglycerides, non-esterified fatty acids and total cholesterol in comparison to the control. In that study, no changes were recorded in blood glucose concentration. Additionally, in rats consuming chia seeds, a reduction of visceral fat level was observed [50]. Comparable research results were obtained by Rossi et al. [51], who reported lower blood concentrations of NEFA and TAG in rats, which were administered feed containing chia seeds, in comparison to the control, in which chia seeds were replaced with corn oil. Additionally, those researchers recorded a lower hepatic TAG level in those animals. That study analysed the effect of chia seed consumption on the activity of selected enzymes involved in the synthesis of fatty acids. It was found that animals fed chia seeds had lower levels of hepatic acetyl-CoA carboxylase (ACC) and fatty acid synthase (FAS) activity in comparison to animals fed a diet rich in corn oil. Additionally, in rats consuming seeds, they recorded a higher activity of carnitine palmitoyltransferase I (CPT-1), participating in beta-oxidation of lipids [51]. 

The effect of chia seed oil consumption on the blood lipid profile was investigated in an experiment conducted by Sierra et al. [52]. They observed that the administration of feed with 10% chia seed oil caused a reduction of total cholesterol, HDL and TG. At the same time the level of LDL increased. However, it needs to be stressed that statistically significant changes were recorded only for TG. Moreover, in rabbits with induced hypercholesterolaemia, a weakened relaxation of aorta vessels in response to acetylcholine (ACh) and a reduced secretion of nitric oxide (NO). The addition of chia seed oil to the diet of rabbits caused increased aorta relaxation, triggered by acetylcholine, and an increased NO release [52].

Fernandez et al. [53] analysed the effect of chia seed consumption on the immune system. Their experimental model was based on the administration of ground chia seeds to rats (at 150 g/kg diet) or chia seed oil (50 g/kg diet) for a period of one month. At the end of the dietary intervention, in both groups, they reported higher concentrations of immunoglobulin E (IgE) in comparison to the control [53].

The effect of consumption of ground chia seeds on blood levels of selected fatty acids in postmenopausal women was assessed in a study conducted by Jin et al. [35]. They showed that a daily intake of 25 g chia seeds during a seven-week dietary intervention caused an increase in blood concentrations of alfa-linoleic acid (by 138%) and eicosapentaenoic acid (by 30%). No differences were observed in the levels of docosapentaenoic or docosahexaenoic acids [35]. In experiments conducted with 12 healthy volunteers Vertommen et al. [54] showed that one-month administration of chia seeds at 50 g daily contributed to a reduction of waist circumference with no simultaneous change in body weight. Moreover, in the participants they recorded a decrease in diastolic blood pressure (DBP) from 66.1 to 61.5 mmHg and a reduced blood triglyceride concentration from 89 to 69 mg/dL [54]. Segura-Campos et al. [55] showed that hydrolysates of chia seed proteins show an activity inhibiting angiotensin convertase (ACE). They observed that the inhibition of activity of the enzyme converting angiotensin is dependent on the duration of hydrolysis. The highest activity was recorded in hydrolysates obtained at 150 min (IC_50_ = 8.86 ug protein/mL), while it was lowest in hydrolysates produced within the time of 90 min (IC_50_ = 44.01 ug/mL) [55]. The inhibitory effect in relation to the activity of angiotensin convertase by fractions of proteins found in chia seeds was confirmed in an experiment performed by Orona-Tamayo et al. [56]. In their study among the analysed protein fractions (albumin, globulin, prolamin, and glutelin), the strongest action inhibiting ACE activity was observed for globulin and albumin. Nieman et al. [57] did not confirm the hypolipemizing action and the effect lowering arterial blood pressure of chia seeds. Consumption of 50 g chia seeds a day by obese men and obese women over a period of 12 weeks caused no changes in values of analysed parameters. No statistically significant differences were recorded in glucose level; concentrations of LDL, HDL and total cholesterol; and blood triglyceride level in comparison to the placebo group. No changes were observed in the concentrations of C-reactive protein (CRP) as well as cytokines: interleukin-6 (IL-6), tumour necrosis factor (TNF-α) and monocyte chemotactic protein (MCP). In that study, a simultaneous increase was found for the blood plasma concentration of alfa-lipoic acid (ALA) in a group of individuals consuming chia seeds. In contrast, no changes were recorded in the concentration of the docosahexaenoic acid (DHA) and eicosapentaenoic acid (EPA) [57]. Similar research results were reported by Nieman et al. [58], who supplemented overweight women aged 49–75 years with a dose of 25 g chia seeds daily for a period of 10 weeks. Among the study participants they reported no changes in the concentration of total cholesterol or blood glucose level. No effect was shown on arterial blood pressure or the level of C-reactive protein. No changes were found in the concentrations of analysed cytokines: interleukin 6 (IL-6), interleukin 8 (IL-8), interleukin 10 (IL-10) or tumour necrosis factor (TNF-α). In contrast, in the blood of female participants following the consumption of ground chia seeds an increase was recorded in the level of alfa-lipoic acid (by 58%) and eicosapentaenoic acid (by 39%). At the same time, no changes were found in ALA and EPA levels in the group of individuals consuming whole chia seeds [58]. 

## 5. Antioxidant and Antimicrobial Activity

Several studies provided evidence for the high antioxidant potential of chia seeds. Sargi et al. [59] showed that chia seeds are capable of deactivating ABTS cation radicals. However, a higher activity was recorded for seeds of brown and golden flax. Those authors also showed that chia seeds exhibit the capacity to scavenge synthetic DPPH radicals and reduce iron ions. Results obtained in both tests indicate a higher antioxidant activity of chia seeds in comparison to flaxseed [59]. Antioxidant activity of chia seeds was also confirmed by Coelho and Salas-Mellado [39]. They showed that extracts from chia seeds are capable of quenching DPPH radicals and they cause their neutralisation by over 70%. Additionally, they showed that these extracts inhibit enzymatic oxidation of guaiacol [39]. In turn, Segura-Campos et al. [55] confirmed that protein hydrolysates from chia seeds are also capable of reducing ABTS cation radicals. They suggested that the tested protein hydrolysates could act as electron donors [55]. A major indicator of antioxidant potential in biological samples is provided by the value of ORAC (Oxygen Radical Absorbance Capacity). It was shown that ORAC of chia seeds is comparable to that of prunes and hazelnuts (Figure 3). Antioxidant activity of compounds contained in chia seeds was also confirmed in the fat emulsion system. Reyes-Caudillo et al. [25] assessed the effect of chia seed extract addition on the degradation rate of beta-carotene in a model system of linoleic acid/beta-carotene in the course of heating at 50 °C. They observed that extracts from chia seeds exhibit antioxidant properties in the model emulsion amounting to 73.5% and 79.3%. They also confirmed the capacity of chia seeds to inhibit lipid peroxidation [25]. 

The antioxidant properties of chia seed may be used, as well as other natural substances (extract of rosemary, extract of tea, gingko biloba extract, phenolic compounds) to protect the lipids and biologically active substances in the oil during storage and use of thermal processes and in new designed food [62,63,64]. In an experiment conducted by Marineli et al. [24], obese rats were fed chia seeds or chia seed oil at 133 and 40 g/kg diet, respectively, for 6 or 12 weeks. Among animals consuming seeds or oil a statistically significant increase was observed in the activity of antioxidant enzymes in blood i.e., that of catalase (CAT), glutathione peroxidase (GPx), glutathione (GSH) and glutathione reductase (GRd), in comparison to the group of animals fed a high-fructose diet with no chia supplementation. No differences were recorded in the concentration of superoxide dismutase (SOD). At the same time, in the case of hepatic antioxidant enzymes they reported an increase in the activity of glutathione reductase and glutathione (only for chia seeds consumed for a period of 12 weeks). In relation to the other enzymes, no differences were observed in their activity. In the same study, after consumption of both chia seeds and chia oil, a reduction was found in blood concentrations of lipid peroxidation biomarkers: 8-isoprostane and malondialdehyde (MDA). No changes were recorded in MDA concentration in livers of the experimental animals [24].

Ayaz et al. [65] demonstrated that consumption of chia seed with plain yogurt as a mid-morning snack induced short-term satiety with no influence on mood states in healthy individuals. A study conducted by Segura-Campos et al. [55] did not confirm any antimicrobial action of protein hydrolysates from chia seeds. No inhibitory effect was observed for protein hydrolysates on the growth of Gram-positive bacteria such as *Klebsiella pneumoniae*, *Staphylococcus aureus*, *Bacillus subtilis*, and *Streptococcus agalactiae* or Gram-negative bacteria such as *Escherichia coli*, *Salmonella typhi* and *Shigella flexneri* [55].

## 6. Application of Chia Seeds in Food Industry

Due to the hydrophilic properties of chia seeds, they are used as substitutes for eggs and fat [43,66,67]. Chia seeds can absorb water in amounts as much as 12-fold greater than their own mass [68]. They provide food with characteristic consistency. At present, chia seeds are used whole, ground and in the form of gel and oil. Gel of chia seeds may be used as a substitute of oil or eggs in baked products. Such application facilitates reduced calorie and fat contents of products. Furthermore, in the case of baked goods, the final product has a greater content of omega-3 acids, which are major biological compounds of great importance for human health. Borneo et al. [69] showed that chia seed gel may replace as much as 25% oil or eggs in cakes. They confirmed that the level of this substitution has an advantageous effect on sensory attributes of the product, such as colour, taste, texture and overall acceptance. However, in the case of replacement of 50–75% oil in dough, an adverse change was observed in density and overall quality of the baked product [69]. As shown by Oliveira et al. [70], flour from chia seeds may also be used to produce pasta as a substitute of wheat flour. In that experiment, they found that pasta prepared with a share of chia flour had a greater nutritive value than the control pasta. It had statistically significantly greater contents of protein, minerals and dietary fibre. It was recorded that pasta with 7.5% wheat flour replaced with chia flour exhibited very good technological properties and received the highest acceptance index in terms of taste [70]. Menga et al. [42] proposed adding chia seeds and mucilage to rice flour for gluten-free fresh pasta. They demonstrated that concentration of 10% of mucilage or chia seeds resulted in nutritious and healthy gluten free pasta with cooking characteristics equivalent to commercial product as confirmed by its firmness.

High acceptance of breads with the addition of chia flour or chia seeds was reported in a study by Coelho and Salas-Mellado [71]. They showed that the introduction of chia flour to breads at 7.8 g/100 g and in the second variant chia seeds at 11.0 g/100 g provided a final product exhibiting a more advantageous ratio of polyunsaturated fatty acids (PUFA) to saturated fatty acids (SAT) than that of the control bread. For traditional bread, the PUFA:SAT ratio was 1.01, while for breads with an addition of chia flour or chia seeds it was 3.1 and 3.9, respectively [71]. Fernandes and Salas-Mellado [72] examined chia mucilage incorporation on the technological quality of breads and pound cakes with a reduced fat content, and showed it to be an effective fat substitute, preserving quality attributes of food products. A positive effect of chia flour addition on the nutritive value and sensory attributes of chips was observed by Coorey et al. [73]. They stated that a 5% substitution of potato flour and rice flour with chia flour is the most advantageous for appearance, colour, aroma, texture, taste and overall acceptability of the final product. In turn, Campos et al. [74] showed that gruel made from chia seeds may be used as a substitute of emulsifiers and stabilisers in the production of ice-cream. However, an adverse change observed in that case of a change of ice-cream colour resulting from the dark colour of chia gel [74]. In turn, Pintado et al. [75] investigated potential substitution of a part or all added fat in frankfurters with chia flour or with an oil in water emulsion prepared by mixing chia flours with water and oil olive. Introduction of chia to frankfurters provided a product enriched with dietary fibre, minerals (potassium, magnesium, calcium, and manganese) as well as mono- and polyunsaturated fatty acids. At the same time, the final product had calorie content reduced by approximately 26% and sensorily acceptable [75]. Ding et al. [43] examined processing properties of chia seed on restructured ham-like products and found that concentration of 1.0% decreased lipid and protein oxidation and improved not only physicochemical and sensorial properties but also added the nutritional value on low fat meat products. The advantageous effect of chia seeds on nutritive value was also shown indirectly when investigating the composition of eggs laid by hens fed a feed with chia seeds. Contents of omega-3 acids (mainly alfa-linoleic acid) increased in their eggs. Addition of chia seeds also caused a reduction of the ratio of omega-6 to omega-3 acid contents as well as the ratio of monounsaturated to polyunsaturated fatty acids [76]. 

It also needs to be stressed that, when chia seeds are not ground to flour, they may be stored for a long time. This results primarily from the husk surrounding the endosperm and secondly from the high contents of compounds with a high antioxidant potential protecting fatty acids against oxidation [77]. At present, in the food industry in various countries worldwide, several products are made either based on chia seeds or fortified with them. These include, e.g., breakfast cereals, cookies, cakes, fruit juices, yogurts, sauces, jams and preserves [78,79]. However, currently both in Poland and in other European countries, their number is relatively low, which results from the binding legal regulations and limits concerning their use in technology. Chia is used in industrial food production as whole seeds, ground or mucilage to increase the nutritional value of the product. There are numerous products with the addition of chia on the market, such as bread, cookies, pasta, ice cream, yogurt, sausages or even ham. It has been found that the industrial use of chia as a fat or egg substitute in food products does not affect significantly their technological or physical properties.

## 7. Placing Chia Seeds on the EU Market—Legal Regulations 

According to the European Union food law, there is a presumption that conventional foodstuffs, in other words food products that have a tradition of use in the EU, may be considered as safe unless new scientific findings indicate otherwise. Safety of foodstuffs that are in some way artificial or earlier unknown for consumers has to be proven before (ex-ante) entering the food market [80]. Despite their natural origin, due to the novelty feature of *Salvia hispanica* seeds with respect to the history of their consumption in Europe, they are classified as novel food. According to Article 3 paragraph 2 point a of Regulation (EU) 2015/2283 of the European Parliament and of the Council of 25 November 2015 on novel foods, amending Regulation (EU) No 1169/2011 of the European Parliament and of the Council and repealing Regulation (EC) No 258/97 of the European Parliament and of the Council and Commission Regulation (EC) No 1852/2001 (OJ L 327, 11.12.2015, p. 1), novel food means any food that was not used for human consumption to a significant degree within the European Union before 15 May 1997, irrespective of the dates of accession of Member States to the Union [81,82]. 

Within the meaning of Regulation (EU) 2015/2283, *Salvia hispanica* (chia) seeds fall into category IV of novel food, namely food consisting of or isolated from, or produced from plants or their parts (Article 3 paragraph 2 point (a) (iv)) [81]. For the purpose of further considerations, it is worth noting that chia seeds are an example of traditional food from third country. Briefly, according to Regulation (EU) 2015/2283 such a food means novel food which is derived from primary production with a history of safe use in a third country. In turn, the safe use in a third country means that safety of the food in question has been confirmed with compositional data and from experience of continued use for at least 25 years in the customary diet of a significant number of people in at least one third country. As already mentioned at the beginning of this article, chia seeds were staple food for peoples inhabiting Central America in pre-Colombian times and currently they are consumed as ingredients or additions to many foodstuffs. 

Since 1 January 2018, date of entry into force of Commission Implementing Regulation (EU) 2017/2470 of 20 December 2017 establishing the Union list of novel foods in accordance with Regulation (EU) 2015/2283 of the European Parliament and of the Council on novel foods [OJ L 351, 30.12.2017, p. 72, as amended], as a rule, novel food products or novel food ingredients can be placed on the EU market or used in production of other foodstuffs by each food business operator on the condition that such products or ingredients in question are included in the Union list of authorised novel foods and all requirements specified in the list are met [82]. Chia seeds and chia oil as such, as well as some food categories with these seeds or oil as ingredients, are included in the list (Table 5).

Therefore, meeting specified conditions on maximum levels and additional labelling, the chia seeds can be used as an ingredient in the following products: bread products; baked products; breakfast cereals; fruit, nut and seed mixes; fruit juice and fruit/vegetable blend beverages; fruit spreads; yoghurt; and sterilised ready to eat meals based on cereals grains, pseudocereals grains and/or pulses. There is no need to undertake any administrative procedure to put the listed food products on the marked. In case of other food products with chia seeds, a food business operator who intends to place such products on the market within the Union should submit a notification of that intention to the Commission. The Commission shall forward the valid notification to the Member States and the European Food Safety Authority. Where no duly reasoned safety objections have been submitted to the Commission, the Commission shall authorise the placing on the market within the Union of the food product concerned and update the Union list.

## 8. Chia Seeds—Future Perspectives

In recent years, there has been an increased interest in chia seeds. This material became the subject of many studies. Perspectives for the use of chia seeds relate to aspects health and also technological. Chia can be part of the new foodstuffs with health-promoting qualities. The seeds are a good source of fibre and can be recommended for diabetes and people with hypercholesterolaemia. Moreover, they can be as supplement in the daily diet because of the high content of omega-3. Studies in vitro and in vivo confirm the health benefits of chia seeds. 

There are numerous proposals on the use of chia in dishes. Chia are characteristic because of the possibility of attractive looking gels and the seeds are used in the form of seed flour or whole seeds. Currently, chia seeds are used in Europe as a component of cereal products, e.g., breakfast cereals, rice crisps, wafers, chips. The use of chia seeds in production of dairy products, fruit and vegetables or meat stuffs have great perspectives. Although there are many gastronomical recipes for food products with chia seeds, at present, their industrial application is hindered by the mentioned legal status of the seed as a novel food ingredient. It means that a food business operator is required to obtain the official premarket approval for each chia product not included in the Union list of authorised novel foods and novel food ingredients, which is established by virtue of Regulation (EU) 2015/2283 of the European Parliament and of the Council of 25 November 2015 on novel foods, amending Regulation (EU) No 1169/2011 of the European Parliament and of the Council and repealing Regulation (EC) No 258/97 of the European Parliament and of the Council and Commission Regulation (EC) No 1852/2001 (OJ 327, 11.12.2015, p.1). 

Chia seeds can be considered as promising component of health-promoting food with increased biological and technological potential. However, similar to other biologically active plants and products of natural origin, chia requires wide-ranging studies on humans to determine its safety, mechanisms of action, as well as efficacy.

## 9. Conclusions

Already thousands of years ago chia seeds were staple food and they were consumed by pre-Columbian peoples living primarily in Central America. In recent years, we have been observing considerable interest in this raw material in relation to its high nutritive value. Chia seeds have high contents of dietary fibre and proteins, rich in many exogenous amino acids. Moreover, chia seeds have high contents of polyunsaturated fatty acids, mainly alfa-linolenic acid, belonging to the group of omega-3 fatty acids. These seeds are also a good source of many minerals and vitamins, as well as bioactive compounds of high antioxidant activity, particularly polyphenols and tocopherols. Current research results indicate many health-promoting properties of chia seeds. These seeds are ascribed a beneficial effect on the improvement of the blood lipid profile. Experiments confirmed their hypotensive, hypoglycaemic, antimicrobial and immunostimulatory effects. Due to the capacity of chia seeds to absorb water and to form gels, they may be used in food technology as a substitute of emulsifiers and stabilisers. In conclusion, chia seeds (*Salvia hispanica*) are a valuable raw material whose technological properties and health-promoting properties can be widely used in the food industry.

## Figures and Tables

**Figure 1 nutrients-11-01242-f001:**
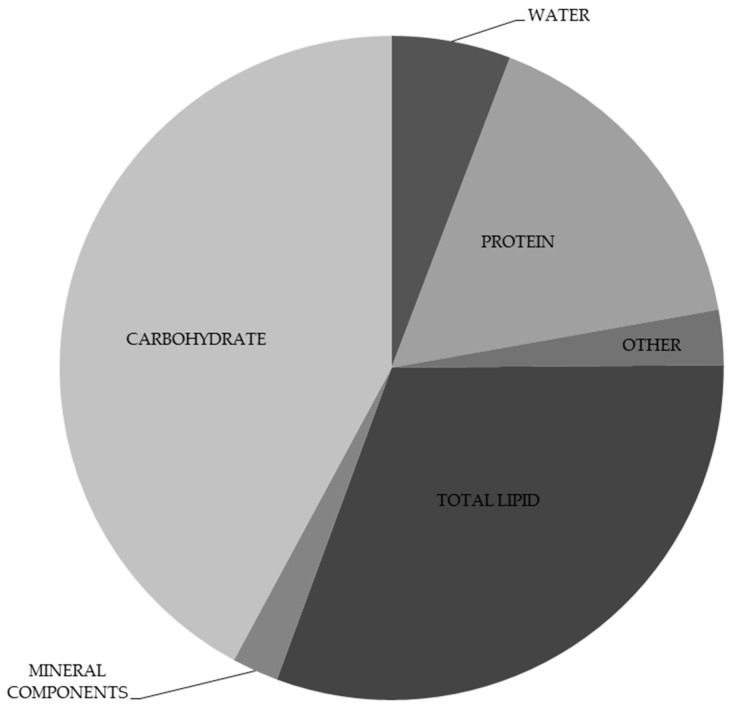
Basic composition of chia seeds [33].

**Figure 2 nutrients-11-01242-f002:**
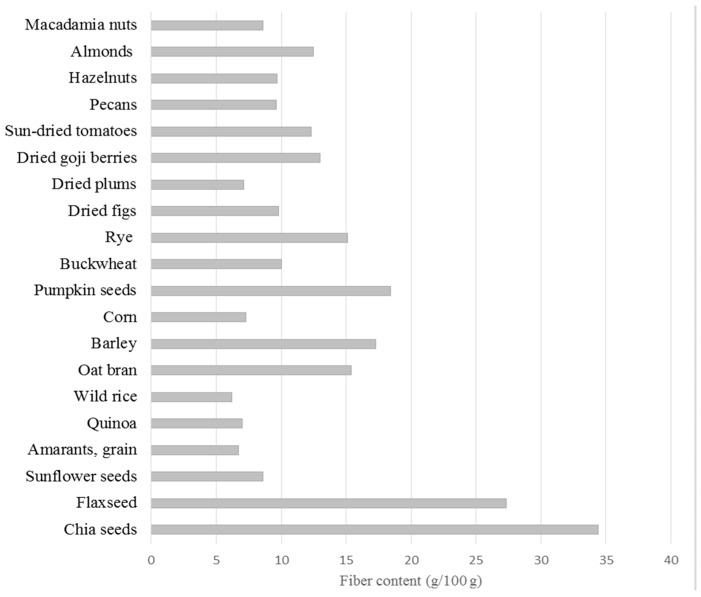
Fibre contents of selected foods [33].

**Figure 3 nutrients-11-01242-f003:**
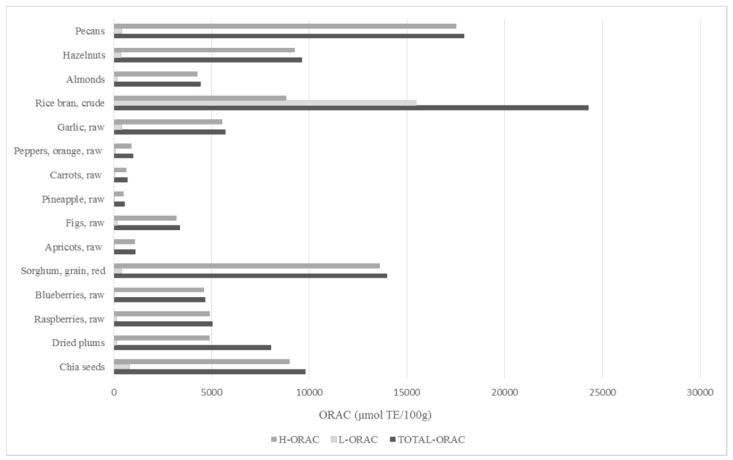
ORAC values for antioxidant activity of selected foods [60,61]. ORAC, Oxygen Radical Absorbance Capacity; H-ORAC, hydrophilic ORAC value for water soluble antioxidants; L-ORAC, lipophilic ORAC value for lipid soluble antioxidants; TE, Trolox equivalents; compilation of results.

**Table 1 nutrients-11-01242-t001:** Nutritional value of chia seeds.

Nutrient	Value			
	USDA [33]		Jin et al. [35]	
Energy	486.0	kcal	562	kcal
Protein	16.5	g/100 g	24.2	g/100 g
Total lipid	30.7		40.2	
Ash	4.8		4.77	
Carbohydrate	42.1		26.9	
Dietary fibre	34.4		30.2	
Calcium	631.0	mg/100 g	456	mg/100 g
Iron	7.7		9.18	
Magnesium	335.0		449	
Phosphorus	860.0		919	
Potassium	407.0		726	
Sodium	16.0		0.26	
Zinc	4.6		6.47	
Copper	0.9		1.86	
Manganese	2.7		3.79	
Vitamin C	1.6			
Thiamine	0.6			
Riboflavin	0.2		n.e.	
Niacin	8.8			
Vitamin E	0.5			
Folate	49.0	µg/100 g	n.e.	µg/100 g

(n.e., not evaluated).

**Table 2 nutrients-11-01242-t002:** A comparison of fatty acid profile of chia and flax seeds (%).

Fatty Acids	Chia		Flax	
	Ciftci et al. [26]	Nitrayova et al. [27]	Ciftci et al. [26]	Nitrayova et al. [27]
**Saturated Fats (SFA)**				
Lauric acid (12:0)	n.e.	0.03	n.e.	0.03
Myristic acid (C14:0)	0.06	0.06	0.07	0.04
Pentadecanoic acid (C15:0)	0.04	n.e.	0.05	n.e.
Palmitic acid (C16:0)	7.1	7.04	5.1	5.39
Margaric acid (C17:0)	0.06	n.e.	0.08	n.e.
Stearic acid (C18:0)	3.24	2.84	3.3	3.17
Arachidic acid (20:0)	0.24	0.02	0.18	0.15
Behenic acid (22:0)	0.08	n.e.	0.14	n.e.
Lignoceric acid (24:0)	0.1	n.e.	0.09	n.e.
**Monounsaturated Fats (MUFA)**				
Palmitoleic acid (C16:1)	0.2	0.03	0.09	0.02
Margaric acid (C17:0)	0.06	n.e.	0.08	n.e
Oleic acid (C18:1 – ω-9)	10.53	7.3	18.1	18.7
Eicosenoic acid (20:1)	0.16	n.e.	0.2	n.e.
**Polyunsaturated Fats**				
Linoleic acid (C18:2 – ω-6)	20.37	18.89	15.3	16.13
Linolenic acid (C18:3 – ω-3)	59.76	63.79	58.2	56.37
Eicosadienoic acid (20:2)	0.08	n.e.	n.e.	n.e.
**Summary**				
SFA	8.65	9.99	7.87	8.78
MUFA	10.95	7.33	18.5	18.72
PUFA	80.4	82.68	73.63	72.5
Ratio n-6/n-3	0.35	0.3	0.27	0.29

(n.e, not evaluated; n-6, omega-6 fatty acids; n-3, omega-3 fatty acids).

**Table 3 nutrients-11-01242-t003:** Amino acid composition of proteins from chia seeds.

Amino Acid	Content (g/100 g)	
	USDA [33]	Nitrayova et al. [27]
**Essential amino acids**		
Arginine	2.14	2.00
Histidine	0.53	0.61
Isoleucine	0.80	0.74
Leucine	1.37	1.42
Lysine	0.97	0.93
Methionine	0.59	0.67
Phenylalanine	1.02	1.6
Threonine	0.71	0.54
Tryptophan	0.44	n/d
Valine	0.95	0.79
**Non-essential amino acids**		
Cystine	0.41	0.42
Tyrosine	0.56	0.61
Alanine	1.04	0.94
Aspartic acid	1.69	1.28
Glutamic acid	3.50	2.87
Glycine	0.94	0.91
Proline	0.78	1.28
Serine	1.05	0.94

**Table 4 nutrients-11-01242-t004:** Polyphenol and isoflavone contents in chia seeds.

	Compound	µg/g Seed	Reference
Polyphenols	Gallic acid	0.05; 11	Jin et al. [35]; Martínez-Cruz and Paredes-López [38]
	Caffeic acid	27; 30.89	Martínez-Cruz and Paredes-López [38]; Coelho and Salas-Mellado [39]
	Chlorogenic acid	4.68	Coelho and Salas-Mellado [39]
	Protocatechuic acid ethyl ester	0.74	
	Ferulic acid	trace	
	Quercetin	0.17	
	Kaempferol	0.013	Jin et al. [35]
	Kaempferol 3-O-glucoside	0.029	
	Epicatechin	0.029	
	Rutin	0.22	
	p-Coumaric acid	0.24	
	Apigenin	0.005	
Isoflavones	Daidzin	6.6	Martínez-Cruz and Paredes-López [38]
	Glycitin	1.4	
	Genistin	3.4	
	Glycitein	0.5	
	Genistein	5.1	

(n.d–not detected).

**Table 5 nutrients-11-01242-t005:** Chia seeds and chia seeds oil as authorised novel food according to Commission Implementing Regulation (EU) 2017/2470 of 20 December 2017 establishing the Union list of novel foods in accordance with Regulation (EU) 2015/2283 of the European Parliament and of the Council on novel foods [81,82].

Authorised Novel Food	Conditions under Which the Novel Food May Be Used		Additional Specific Labelling Requirements
**Chia seeds (*Salvia hispanica*)**	**Specified food category**	**Maximum levels**	1. The designation of the novel food on the labelling of the foodstuffs containing it shall be “Chia seeds (*Salvia hispanica*)” 2. Pre-packaged Chia (*Salvia hispanica*) seeds shall carry additional labelling to inform the consumer that the daily intake is no more than 15 g
	Bread products	5% (whole or ground chia seeds)	
	Baked products	10% whole chia seeds	
	Breakfast cereals	10% whole chia seeds	
	Fruits, nut and seed mixes	10% whole chia seeds	
	Fruit juice and fruit/vegetable blend beverages	15 g/day for addition of whole, mashed or ground chia seeds	
	Pre-packaged Chia seed as such	15 g/day whole chia seeds	
	Fruit spreads	1% whole chia seeds	
	Yoghurt	1,3 g whole chia seeds per 100 g of yoghurt or 4,3 g whole chia seeds per 330 g of yoghurt (portion)	
	Sterilised ready to eat meals based on cereal grains, pseudocereals grains and/or pulses	5% whole chia seeds	
**Chia oil from *Salvia hispanica***	**Specified food category**	**Maximum levels**	The designation of the novel food on the labelling of the foodstuffs containing it shall be “Chia oil (*Salvia hispanica*)”
	Fats and oils	10%	
	Pure chia oil	2 g/day	
	Food supplements as defined in Directive 2002/46/EC	2 g/day	

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
