# Peer review of "The Chemical Composition and Nutritional Value of Chia Seeds—Current State of Knowledge"

_nutrients, 2019, doi:10.3390/nu11061242_

Round 1

Reviewer 1 Report

The paper entitled "Chia seeds the nutritional value of an ancient grain – current state of knowledge" is a very interesting review, it certainly can be published in Nutrients after correction and clarification of the following details.

1)      Title: I suggest modifying it. Maybe it sounds better: The chemical composition and nutritional value of Chia seeds. Current state of knowledge. I think that “an ancient grant” is not relevant for the title.

2)      The abstract is short and clear. I suggest to include something more information in the abstract.

3)      The main topic of the review is the Chia seeds.  So, I suggest starting the introduction by describing the Chia.

4)      Lines 41-43. It is not scientific information. I suggest removing it.

5)      Table 1: Supress the “units” column and add the units in the data of the rest of columns. E.g. 486.0 kcal instead of 486.0

6)      Both Figure 1 and Figure 2 contain information about different foods. Although these figures are interesting, I suggest include figures focusing only on Chia. For example, it could be very interesting to add a figure (like circular graph) detailing the composition of the Chia. Figure 1 and 2 may also be included (maybe better like a supplementary material).

7)      Table 4 contains a lot of n.e and n.d values. I suggest modifying it to show only the relevant information

8)      The subsection “Health-promoting properties” is difficult to read. Describing each study for separate is confusing. I suggest redoing this section. It may be more clear starting with in vivo studies and after with humans studies. On the other hand, another choice could be describe each property for separate. Are there in vitro studies related to this section? It could be interesting to include them.

9)      Sections 6 and 7 are very clear and interesting for the readers. It is a good point of the review.

10)   The conclusions are mentioned in the section 8 and 9. Please, not repeat the information in both section.

Author Response

Poznan University of Life Sciences                                                                                        Poznan 23.05.2019

Faculty of Food Science and Nutrition

Wojska Polskiego 31

60-637 Poznan, Poland

tel. (+48 61) 848 7096

fax. (+48 61) 848 7146, 848 7145

e-mail: Food_Sci@up.poznan.pl

 Editorial Board

Nutrients

Dear Editor,

Please find below detailed references to the Reviewer's comments.

Journal: Nutrients

Manuscript ID: nutrients-510001

Title: The chemical composition and nutritional value of Chia seeds – current state of knowledge.

Comments from the Reviewers:

Reviewer 1

Comments and Suggestions for Authors

The paper entitled "Chia seeds the nutritional value of an ancient grain – current state of knowledge" is a very interesting review, it certainly can be published in Nutrients after correction and clarification of the following details.

1)      Title: I suggest modifying it. Maybe it sounds better: The chemical composition and nutritional value of Chia seeds. Current state of knowledge. I think that “an ancient grant” is not relevant for the title.

Response to the query:

The title has been modified according to Reviewer’s comment and is as follows: “The chemical composition and nutritional value of Chia seeds - current state of knowledge”.

2)      The abstract is short and clear. I suggest to include something more information in the abstract.

Response to the query:

The abstract has been supplemented with the sentence: “Research indicates that components of chia seeds are ascribed a beneficial effect on the improvement of the blood lipid profile, through their hypotensive, hypoglycemic, antimicrobial and immunostimulatory effects.”

3)      The main topic of the review is the Chia seeds.  So, I suggest starting the introduction by describing the Chia.

Response to the query:

The authors agree with the Reviewer's suggestion that the publication concerns chia and that the introduction should apply directly to it. The intention of the authors was to briefly introduce the reader to the importance of nutrition and appropriate configuration of a diet rich in bioactive compounds. Afterwards, it was pointed out that chia seeds are a raw material that meets the requirements of the so-called superfood and has promising pro-health properties that should be investigated. The detailed characteristics of chia seeds are presented in the subsequent chapter, which is why it was considered that broadening of the preliminary characterization of chia may be too soon.

4)      Lines 41-43. It is not scientific information. I suggest removing it.

Response to the query:

The sentence has been removed

5)      Table 1: Supress the “units” column and add the units in the data of the rest of columns. E.g. 486.0 kcal instead of 486.0

Response to the query:

The Unit column has been removed and modified according to Reviewers suggestion.

6)      Both Figure 1 and Figure 2 contain information about different foods. Although these figures are interesting, I suggest include figures focusing only on Chia. For example, it could be very interesting to add a figure (like circular graph) detailing the composition of the Chia. Figure 1 and 2 may also be included (maybe better like a supplementary material).

Response to the query:

Placing the two figures in the publication was aimed at comparison with other products present in the diet of the average consumer. Therefore, the authors wanted to ask for the possibility of leaving these drawings at work. Corrected text includes a figure with the basic composition of chia seeds in accordance with the Reviewer’s request.

7)      Table 4 contains a lot of n.e and n.d values. I suggest modifying it to show only the relevant information

Response to the query:

The table has been modified according to Reviewer’s comment.

8)      The subsection “Health-promoting properties” is difficult to read. Describing each study for separate is confusing. I suggest redoing this section. It may be more clear starting with in vivo studies and after with humans studies. On the other hand, another choice could be describe each property for separate. Are there in vitro studies related to this section? It could be interesting to include them.

Response to the query:

The subsection was re-edited in accordance with the Reviewer's suggestions. In vitro tests have been presented in the chapter on antioxidant activity.

9)      Sections 6 and 7 are very clear and interesting for the readers. It is a good point of the review.

Response to the query:

The authors appreciate the recommendation.

10)   The conclusions are mentioned in the section 8 and 9. Please, not repeat the information in both section.

Response to the query:

The repeated phrases from both sections were eliminated, the sentence on EU recommendations was removed from section 9.

Thank you for the thorough and detailed review of our publication.

I would be grateful if the manuscripts could be reviewed and reconsidered for publication in Nutrients.

Yours sincerely

Anna Gramza-Michalowska, prof., DSc, PhD

Reviewer 2 Report

Perhaps a slight change in the title to read:

"Chia seeds: the nutritional value of an ancient grain - current state of knowledge"

The topic will be of interest to the general readers. My suggestions to improve the paper are highlighted in yellow within the manuscript (attached).

L201-207: The sentence need to be clearly connected with the previous one, anti-microbial action was thrown in completely out of the blues. Perhaps it can be considered that the authors combine this with antioxidant activity and change the title of section 5 to "Antioxidant and antimicrobial activities".

L 300-302: Here the authors may write a bit more on industrial food fortification with chia seeds, oil and flour in several food products.

L397-398: The concluding statement need to be revised.

As a review paper on the nutritional value of an ancient grain and its current state of knowledge. This will require some emphasis on the nutritional value in humans - to be well captured, especially immediately after section 3 where the authors refer to the chemcal composition of chia seeds.

Kindly revise the grammar and use of English in the text, some of these I have attempted to revise. A professional English reviewer is highly recommended.

Author Response

Poznan University of Life Sciences                                                                                        Poznan 23.05.2019

Faculty of Food Science and Nutrition

Wojska Polskiego 31

60-637 Poznan, Poland

tel. (+48 61) 848 7096

fax. (+48 61) 848 7146, 848 7145

e-mail: Food_Sci@up.poznan.pl

 Editorial Board

Nutrients

Dear Editor,

Please find below detailed references to the Reviewer's comments.

Journal: Nutrients

Manuscript ID: nutrients-510001

Title: The chemical composition and nutritional value of Chia seeds – current state of knowledge.

Comments from the Reviewers:

Reviewer 2

1)         Perhaps a slight change in the title to read:

"Chia seeds: the nutritional value of an ancient grain - current state of knowledge"

Response to the query:

The paper’s title has been changed according to Reviewer’s suggestion.

2)         The topic will be of interest to the general readers. My suggestions to improve the paper are highlighted in yellow within the manuscript (attached).

Response to the query:

The paper has been carefully revised as suggested by the Reviewer.

3)         L201-207: The sentence need to be clearly connected with the previous one, anti-microbial action was thrown in completely out of the blues. Perhaps it can be considered that the authors combine this with antioxidant activity and change the title of section 5 to "Antioxidant and antimicrobial activities".

Response to the query:

The part of the manuscript has been modified according to Reviewer’s comment.

4)         L 300-302: Here the authors may write a bit more on industrial food fortification with chia seeds, oil and flour in several food products.

Response to the query:

The part of the manuscript has been modified according to Reviewer’s comment: “Chia is used in industrial food production as whole seeds, ground or mucilage in order to increase the nutritional value of the product. There are numerous products with the addition of chia on the market, such as bread, cookies, pasta, ice cream, yogurt, sausages or even ham. It has been found that the industrial use of chia as a fat or egg substitute in food products does not affect significantly their technological or physical properties.”

5)         L397-398: The concluding statement need to be revised.

Response to the query:

The concluding statement has been revised as follows: “In conclusion, chia seeds (Salvia hispanica) are a valuable raw material whose technological properties and health-promoting properties can be widely used in the food industry”.

6)         As a review paper on the nutritional value of an ancient grain and its current state of knowledge. This will require some emphasis on the nutritional value in humans - to be well captured, especially immediately after section 3 where the authors refer to the chemical composition of chia seeds.

Response to the query:

According to the reviewer's suggestion, a part of the influence of chia ingredients on the human body was re-edited after the third chapter.

7)         Kindly revise the grammar and use of English in the text, some of these I have attempted to revise. A professional English reviewer is highly recommended.

Response to the query:

The manuscript English language was checked by the translation service according to Reviewer’s suggestions.

Thank you for the thorough and detailed review of our publication.

I would be grateful if the manuscripts could be reviewed and reconsidered for publication in Nutrients.

Yours sincerely

Anna Gramza-Michalowska, prof., DSc, PhD

Round 2

Reviewer 1 Report

Almost all of my suggestions have been included so I suggest accepting the work.

Minor comments: The Table 4 should be deleted or remaded in another way. A table with so few values does not fit well.

Author Response

Poznan University of Life Sciences                                                                               Poznan 24.05.2019

Faculty of Food Science and Nutrition

Wojska Polskiego 31

60-637 Poznan, Poland

tel. (+48 61) 848 7096

fax. (+48 61) 848 7146, 848 7145

e-mail: Food_Sci@up.poznan.pl

 Editorial Board

Nutrients

Dear Editor,

Please find below detailed references to the Reviewer's comments.

Journal: Nutrients

Manuscript ID: nutrients-510001

Title: The chemical composition and nutritional value of Chia seeds – current state of knowledge.

Comments from the Reviewers:

Reviewer 1

Comments and Suggestions for Authors

Almost all of my suggestions have been included so I suggest accepting the work.

Minor comments: The Table 4 should be deleted or remaded in another way. A table with so few values does not fit well.

Response to the query:

The Table 4 The table has been re-edited and its layout has been changed as suggested by the reviewer.

Thank you for the thorough and detailed review of our publication.

I would be grateful if the manuscripts could be reviewed and reconsidered for publication in Nutrients.

 Yours sincerely

Anna Gramza-Michalowska, prof., DSc, PhD
